# Analysis and Verification of the Method of Improving Inductance by Magnetic Endcaps in Slotless Permanent Magnet Motor

Chenglong Chu and Yunkai Huang *

School of Electrical Engineering, Southeast University, Nanjing 210096, China; chucl@seu.edu.cn
* Correspondence: huangyk@seu.edu.cn

**Abstract:** The slotless permanent magnet motor (SPMM) has low phase inductance due to the larger physical air gap, which will adversely affect motor control and current harmonics. In this paper, the method of forming an extra magnetic circuit by endcap around the outer stator is proposed. The advantage of this method is that a restrained flux path is formed without increasing the motor structure and cost, and the inductance of the motor is effectively improved without causing a significant decrease in torque. The preliminary simulation analysis and corresponding experimental content are carried out. The experimental results and the simulation content showed good consistency, which verified the correctness of the theory and simulation analysis.

**Keywords:** current harmonics; inductance; slotless permanent magnet motor (SPMM); magnetic circuit

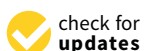



## 1. Introduction

Permanent magnet synchronous motors (PMSM) are increasingly used in electric and hybrid vehicles, solar aircraft, flying vehicles, compressor pumps and other applications due to their high efficiency, high power density and good control performance [1,2]. With the rapid development of electrification in transportation industry, motors are required to have a higher power density [3].

For the traditional tooth-slot structure, the main flux is confined to the stator teeth so that the motor has stronger torque output capacity. However, in the case of high power density, the increase of operating frequency and electromagnetic load makes the tooth easy to appear magnetic saturation and obvious loss [4]. In addition, the slotted structure will naturally lead to the groove torque of the permanent magnet motor, affecting the motor control performance and operation performance. In contrast, the slotless stator structure can eliminate the cogging torque in the permanent magnet synchronous motor and significantly reduce the torque ripple of the motor. The slotless motor with high pole pairs makes the stator core thinner and further improves the power density of the motor.

However, the larger air gap leads to a smaller inductance, which not only affects the drive performance of motors, but also generates obvious current harmonic and a large amount of eddy current loss on the rotor [5,6]. Therefore, how to effectively increase the inductance and suppress current harmonics is the key points in the design of SPMM. In [3], a stator embedded inductor integrated with the torque producing machine windings is used to reduce the PWM induced current ripple, short circuit current, and current ripple induced core losses. However, this method makes the stator core structure more complicated and the extra windings need to be inserted into slots inside the core, which adds to the difficulty and cost of manufacturing. In addition, both the more complex stator configuration and the additional windings increase the weight of the motor, resulting in a decrease in power density.

The outer slot-tooth structure in stator iron is used to increase leakage inductance and solve the problem of low inductance of high speed permanent magnet synchronous motor

in [7]. However, this structure will increase iron consumption, and make the weight and volume of the motor rise significantly. Meanwhile, the structure also significantly increased the difficulty of winding. More importantly, the significant increase in flux leakage reduces the flux linkage interaction with the rotor, resulting in a significant reduction in torque.

In addition, methods such as adjusting DC bus voltage, connecting a three-phase LC filter circuit between windings and inverter, a multi-level inverter circuit [8,9], and increasing switching frequency [10] are usually used to suppress current harmonics. However, the above methods will either cause motors and its drive system more complex and expensive or increasing the difficulty of motor control [11].

This paper presents a method for increasing inductance of slotless motor. In this method, an additional magnetic circuit is formed on the outer side of the stator by magnetic endcaps, which can effectively increase the inductance without series inductor and by designing additional stator structure. At the same time, there will be no significant torque drop. The principle and simulation of this method will be described in detail in the following content, and finally verified by prototype test.

## 2. Motor Design Parameters and the Influence of Endcap Materials

### 2.1. Motor Design Parameters

The prototype of this paper uses a slotless stator core with toroidal windings, and its 3D structure diagram and cross-sections are shown in Figures 1 and 2. Table 1 lists the relevant design parameters of SPMM. The SPMM with 12-teeth/10-poles, and stator core has a thickness of 9 mm. The stator core is made of 0.2 mm WTG200 laminated, and the skeleton is processed at the outer diameter. The outside of stator extends 12 stator skeletons to separate adjacent coils and install endcaps. The two endcaps are fixed by multiple screws. In the rotor, carbon fiber sleeve is utilized to effectively protect magnets attached to rotor surface. Toroidal windings form multiple ring structures around the inner wall and the outer side of the stator core, which can greatly reduce the length of end windings and axial installation size of SPMM [12].

Generally, inner windings can be fixed with 3D printed framework and epoxy resin filling, so as to avoid friction between inner windings and the rotor due to inner windings' own quality and the interaction between armature magnetic field and permanent magnetic field during operation. The cured epoxy resin can not only fix windings, but also effectively fill the gap between conductors, reduce the equivalent thermal resistance inside windings, and improve the heat dissipation capacity.

Rotor eddy current loss of fractional slot concentrated winding (FSCW) motor mainly comes from armature magnetic potential harmonics, which is distinct from integer slot distributed winding (ISDW) motor. Therefore, the use of low-conductivity materials such as carbon fiber sleeve and bonded magnets can effectively reduce rotor eddy current loss [13]. In addition, the lower order MMF harmonics in FSCW motor can penetrate into rotor yoke, resulting in greater eddy current loss [14]. When the Halbach array is used, rotor yoke can be made of non-magnetic materials, which further reduces eddy current loss.

**Table 1.** DESIGN PARAMETERS OF THE SPMM-GRW.

| Number of poles | 10 | Number of slots | 12 |
|---|---|---|---|
| Rated power (kW) | 3.0 | Rated speed (r/min) | 3000 |
| PM thickness (mm) | 18 | Winding type | Gramme |
| Turns per coil | 25 | Diameter of stator bore (mm) | 132 |
| Stator stack length (mm) | 180 | Physical air gap (mm) | 2 |

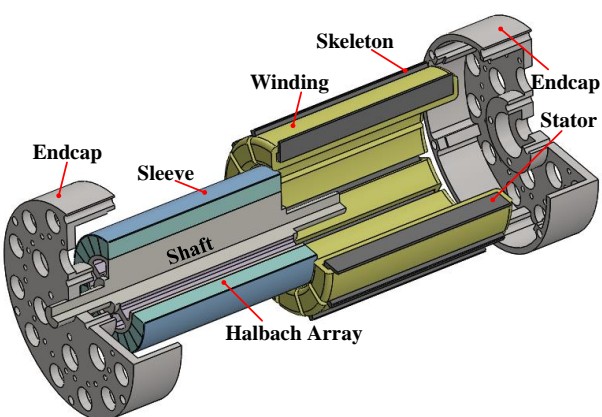

**Figure 1.** The 3D structure diagram of the SPMM-GRW.

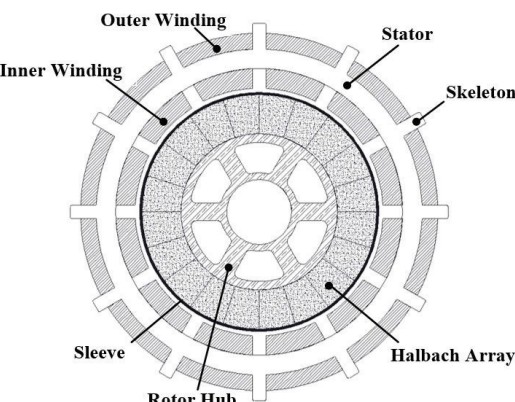

**Figure 2.** The cross-sections of the SPMM-GRW.

*2.2. Proposed Method for Increasing Inductance*

In slotless motor, the air gap of the motor is much larger than that of the slotted motor because there is no tooth structure. The inductance of winding is inversely proportional to the air gap of the motor, so the inductance of slotless motor is usually small. At present, the existing methods such as adjusting DC bus voltage, connecting a three-phase LC filter circuit between windings and inverter, a multi-level inverter circuit, and increasing switching frequency and other methods have certain limitations. Therefore, this paper presents a method for simple and effective increasing inductance of slotless motor.

Refer to Figure 1 for illustration, the installation of SPMM requires skeletons, endcaps and screws to cooperate with each other. Endcaps are tightly connected to stator. In addition to directly affecting the assembly of SPMM, the materials can also affect the performance of the motor. Generally, the endcaps usually chooses aluminum alloy material, because of its lower price and lighter weight. This paper proposes the method to replace non-magnetic materials like aluminum alloy with magnetic materials which can significantly increase the inductance of slotless motor.

For the case of using non-magnetic material endcaps, when windings are supplied with an alternating current which the effective value is $I$, the flux linkage $\psi_{coil}$ through entire windings. Among them, the schematic diagram of flux linkage is shown on the left in Figure 3 and the inductance of the winding is

$$L = \frac{\psi_{coil}}{I} \tag{1}$$

And if the endcaps adopt magnetic material, there will be two magnetic circuits. One of them is the magnetic circuit surrounding the inner winding passes through the stator and interacts with the rotor magnetic field to realize electromechanical energy conversion,

the flux linkage is $\psi_m$ The additional one is the magnetic circuit surrounding the outer windings protrudes through the skeletons of the stator and the endcaps, the flux linkage is $\psi_{add}$ which does not interact with the rotor's magnetic field. That is, in addition to self-inductance, the additional stator magnetic circuit produces additional inductance. The total inductance of the winding is increased by series connection of two inductances, as shown in (2).

$$L = \frac{\psi_m + \psi_{add}}{I} = L_m + L_{add} \tag{2}$$

Since the contact area between the endcap and the skeleton is limited, $\psi_m$ will be slightly smaller than the $\psi_{coil}$. When the number of turns is constant, the inductance is inversely proportional to the magnetic resistance of the magnetic circuit. Therefore, although the proportion of $\psi_{add}$ is not high, the distance between the endcap and the skeleton is extremely small, and the reluctance of the entire magnetic circuit is very small, resulting in significant additional inductance.

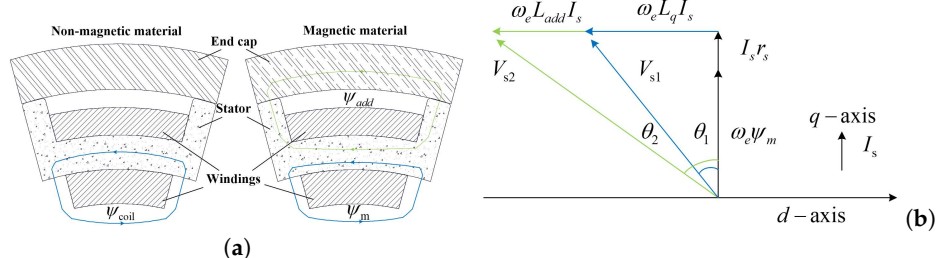

**Figure 3.** Schematic diagram of flux linkage of different materials and vector diagram of different materials. (**a**) Schematic diagram (**b**) vector diagram.

As the stator forms an additional magnetic circuit, the flux through the coil and rotor side is slightly reduced, resulting in a slight decrease in output torque. In the slotless-Halbach machine, the d-axis current is set to zero to maximize the torque output. The phase voltage during maximum torque per ampere (MTPA) control is [11]

$$V_s = \sqrt{(\omega_e \psi_m + r_s I_s)^2 + (\omega_e L_d I_s)^2} \tag{3}$$

where $V_s$ is the terminal voltage, $\omega_e$ is the angular frequency, $r_s$ is the phase resistance, $\psi_m$ is the magnet flux linkage, $I_s$ is the phase peak current, and $L_d$ is the d-axis inductance. It can be seen from (3), when the stator forms an additional magnetic circuit, the inductance of winding increases, $L_d$ increases accordingly. And the terminal voltage of each phase increases.

This conclusion can also be seen from the vector diagram, as shown in Figure 3b. Where, $V_{s1}$ and $V_{s2}$ are the terminal voltage values of the non-magnetic endcaps and the magnetic endcaps respectively, and the included angles with the current $I_s$ are $\theta_1$ and $\theta_2$ respectively. It is obvious that when the magnetic endcaps are used, the terminal voltage and the angle between the terminal voltage and the current are slightly larger. Because the power factor of the motor is proportional to the cosine of the absolute value of the angle between voltage and current, the power density will decrease slightly when the magnetic endcaps are used.

## 3. The Preliminary Simulation Verification

The magnetic flux distribution of the stator with different materials is shown in Figure 4. When endcaps are made of non-magnetic material such as aluminum alloy (abbreviated as AL), the flux linkage in endcaps is basically zero, and the flux linkage of winding passes through the inner side of stator and air gap, interacts with rotor magnetic field to realize electromechanical energy conversion. Whereas, magnetic material such as S45C is used for preliminary validation because it is easy to obtain and process. In addition

to the inner flux linkage interacting with rotor magnetic field in windings, there is also the outer magnetic circuit passing through skeletons and endcaps, and the phase inductance has increased. In order to obtain accurate phase inductance, the LCR method is employed in 3D FEA.

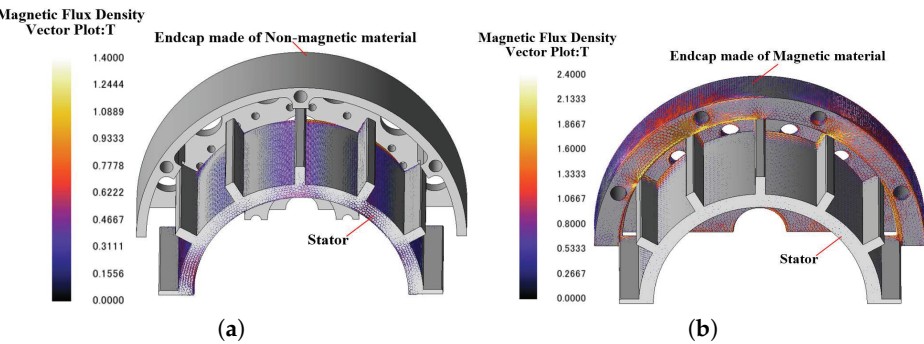

**Figure 4.** Magnetic flux distributions when installing end caps made of different materials. (**a**) Non-magnetic material. (**b**) Magnetic material.

The LCR method is a simulation of actual motor parameters measured by LCR meter. In 3D-FEA, the rotation speed is set to 0 and three-phase windings is suspended, and one phase is set to excitation with the same amplitude and frequency as the LCR meter. Calculate the real and imaginary parts of voltage and current waveforms through FFT transformation, and use following formula to calculate phase inductance. In AC state, the relationship between voltage, current and parameters of each phase windings is

$$\dot{V} = (R + j\omega L)\dot{I} \tag{4}$$

where, $\dot{V}$ and $\dot{I}$ respectively represent the voltage and current of each phase winding, which are both complex numbers; $R$ and $L$ represent the resistance and inductance of each phase winding respectively; $\omega$ is the angular frequency; $j$ is an imaginary unit. In AC state, the relationship between voltage, current and parameters of each phase windings is

$$\dot{V} = (R + j\omega L)\dot{I} \tag{5}$$

where, $\dot{V}$ and $\dot{I}$ respectively represent the voltage and current of each phase winding, which are both complex numbers; $R$ and $L$ represent the resistance and inductance of each phase winding respectively; $\omega$ is the angular frequency; $j$ is an imaginary unit.

$$(R + j\omega L) = \frac{\dot{V}\dot{I}^*}{\left|\dot{I}\right|^2} = \frac{\dot{V}_{re}\dot{I}_{re} + \dot{V}_{im}\dot{I}_{im}}{\left|\dot{I}\right|^2} + j\frac{\dot{V}_{im}\dot{I}_{re} - \dot{V}_{re}\dot{I}_{im}}{\left|\dot{I}\right|^2} \tag{6}$$

and parameter $\dot{I}^*$ is the conjugate complex number of $\dot{I}$, the resistance and inductance parameters are calculated as follows

$$R = \frac{\dot{V}_{re}\dot{I}_{re} + \dot{V}_{im}\dot{I}_{im}}{\left|\dot{I}\right|^2}; L = \frac{\dot{V}_{im}\dot{I}_{re} - \dot{V}_{re}\dot{I}_{im}}{\omega\left|\dot{I}\right|^2} \tag{7}$$

The phase inductance at different rotor position (step: 6 degrees) obtained by the LCR method when using different materials are shown in Figure 5. It can be seen that the phase inductance of the conductive endcaps is 1.13 mH, while that of the non-conductive material is only 0.62 mH, which is about 70% higher.

In addition, the alternating magnetic field generated by stator windings will produce higher eddy current loss in magnetic end caps. The Joule loss density when using different materials is shown in Figure 6. The eddy current loss in magnetic end caps is 151.2 W, while eddy current loss in non-magnetic end caps is only 15.8 W.

In addition, as shown in Figure 7, when the motor adopts magnetic endcaps in rated operation, electromagnetic torque is reduced by about 3.8% compared to when non-magnetic materials are used. The terminal voltage with the magnetic endcaps is increased by about 20% and it is slightly ahead of that of the non-magnetic endcaps, and the power factor is reduced by about 4%. These electromagnetic performance changes of the motor are consistent with the analysis in the previous section.

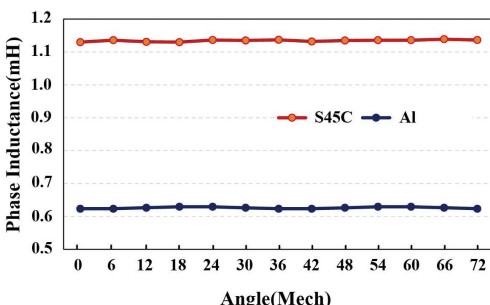

**Figure 5.** Calculated phase inductance at different rotor position.

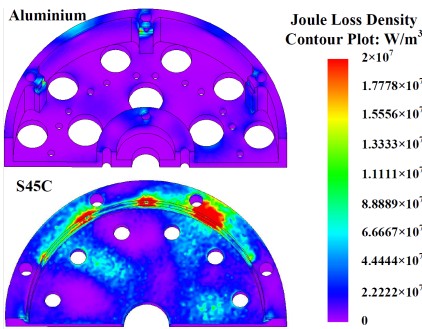

**Figure 6.** The Joule loss density of end caps made of different material.

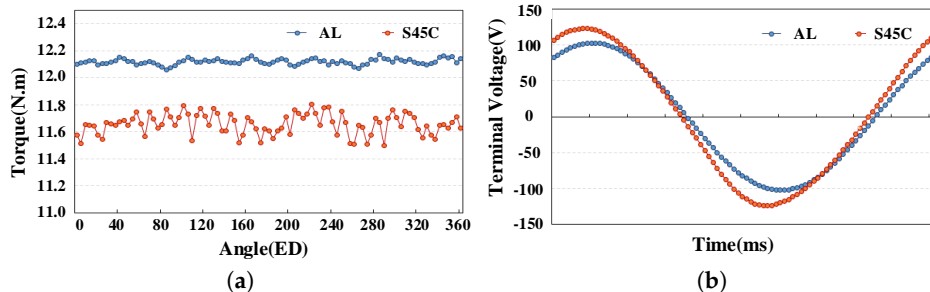

**Figure 7.** Electromagnetic torque and terminal voltage waveform of the motor during rated operation. (**a**) Electromagnetic torque. (**b**) Terminal voltage.

## 4. Experimental Verification

### 4.1. Inductance Measurement

A prototype is manufactured and a series of prototype tests are carried out to verify the results of simulation and analysis. In order to determine the influence of the material of endcaps on phase inductance, two pairs of endcaps of different materials are made and phase inductances at different rotor positions are measured with an LCR meter, as shown in Figure 8a.

Measured inductances against rotor position are plotted in Figure 8. In order to measure more accurate inductance parameters, the inductance at every 1 degree of the rotor is tested. The mean values of phase inductance measured with different end caps are 1.11 mH and 0.62 mH, respectively. The LCR meter is used to measure the values of phase inductance at different rotor position, which is basically consistent with calculated

results in Figure 8b. For the end caps made of two different materials, aluminum end caps have certain advantages in quality due to the low density of aluminum. Specifically, the individual weight of aluminum end cover is 1.1 kg, while the individual weight of S45C end cover is 3.2 kg.

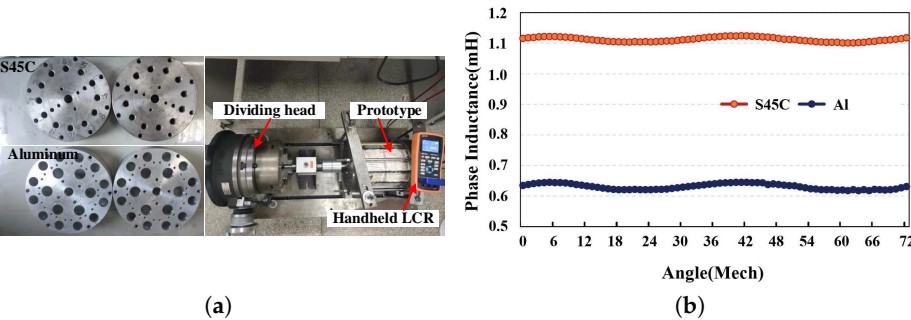

| (a) | (b) |
|-----|-----|

**Figure 8.** Different end caps and measured phase inductance. (**a**) End caps. (**b**) Measured phase inductance.

### 4.2. Open Circuit Experiment

The back-EMF test platform is shown in Figure 9 and the terminal voltage of each phase winding are displayed on oscilloscope. The motor under test is driven by a servo motor to 3000 r/min. When the prototype is assembled with endcaps of different materials, the calculation of back EMF and the comparison of experimental waveforms are shown in Figure 10.

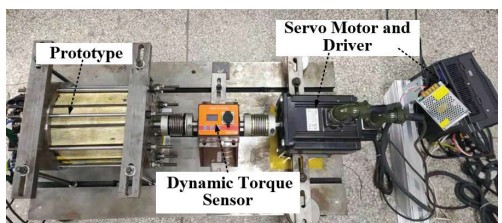

**Figure 9.** Open circuit experimental setups.

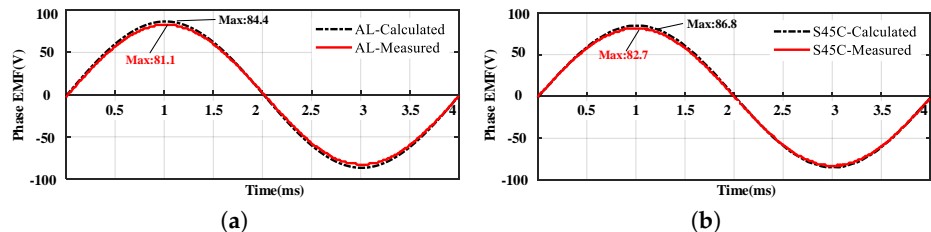

| (a) | (b) |
|-----|-----|

**Figure 10.** Calculated and measured back EMFs of different endcaps at 3000 r/min. (**a**) Aluminum. (**b**) S45C.

It can be seen that the back EMF of AL-endcaps is slightly higher than that of S45C. Moreover, the measured and calculated values under different endcaps materials are very consistent, and the experimental values are slightly lower than the calculated values but the errors are all within 5%.

The measurement of static torque is to verify the torque output capability of the prototype. In static torque experiment, windings are connected to a DC power supply, but it should be noted that the current in one phase is twice that of the other two, and the direction is opposite.

The data measured by torque sensor are displayed on an oscilloscope via a voltage probe, and the rotor position is usually controlled by rotating indexing head. In this experiment, the experimental platform is shown in Figure 11a, and the rotor position is

controlled by a servo motor connected to a reducer (reduction ratio 5:1). Extremely low speed (0.2 r/min) will not affect DC power supply, and the test content can be completed more conveniently under the premise of test accuracy. The test results are shown in Figure 12, and show a good agreement with calculated results , where the maximum error is 6.4%.

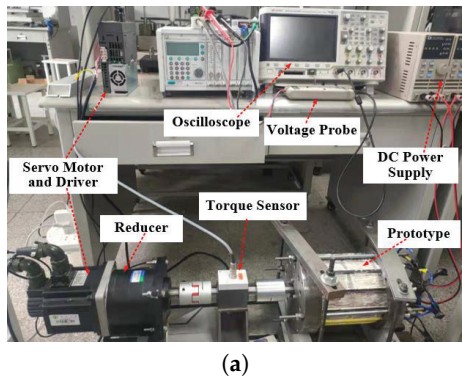 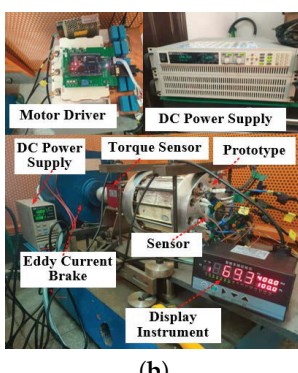

(**a**) (**b**)

**Figure 11.** Different end caps and measured phase inductance. (**a**) End caps. (**b**) Measured phase inductance.

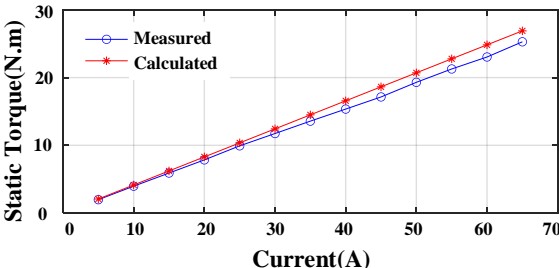

**Figure 12.** Measured and Calculated static torques.

### *4.3. On Load Experiment*

The prototype load test platform is shown in Figure 11b. The large DC power supply in the picture is provided for the windings of SPMM through motor driver. The motor driver adopts SPWM modulation mode, and the switching frequency is 10 kHz. The eddy current brake is used as the load of the prototype and the output torque of the motor is modified by controlling the current of the connected small DC power supply.

In order to verify the influence of endcaps material on load current, two experiments are carried out. Among them, the rotation speed is 1000 r/min, the current of external power supply of the eddy current brake is constant 0.49 A, the output torque of the motor is 6.85 N.m, and load current waveforms and spectra are shown in Figure 13. For better identification and comparison, the data in "-AL" waveform are reversed.

Through the previous analysis, it is obvious that the phase inductance is larger when the magnetic endcaps are used, which can suppress current harmonics. The current waveform in Figure 13 can well verify this conclusion. The THD of load current using magnetic endcaps(black curve) is only 2.62%, while load currently using non-magnetic endcaps is 4.27%, and the total harmonic distortion (THD) of the current waveform decreasing by 38%.

In addition, the analysis in previous section shows that under the same current excitation, the output torque of magnetic endcaps is slightly smaller. In other words, when the prototype is required to output the same load torque, the load current of the motor with magnetic endcaps is higher, which corresponds to the result in spectra. In other words, when the prototype needs to output the same load torque, the load current of the magnetic endcaps will be slightly higher. According to the experimental results, when S45C is used as the material of the end cap, the fundamental current amplitude is 20.64 A,

while the fundamental current amplitude of aluminum end cap is 19.5 A. The experimental results fully verify the correctness of the analysis.

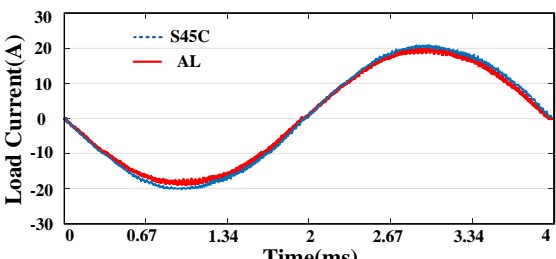

**Figure 13.** The comparision of current waveform.

## 5. Conclusions

This paper proposes a method to increase the inductance of the motor by forming an additional magnetic circuit on the periphery of the stator with magnetic endcaps. Through theoretical derivation, simulation analysis and experimental verification, it is determined that in a preliminary demonstration of this method, an increase of over 70% in phase inductance is achieved, and the THD of the current waveform decreasing by 40%. However, the end cap made of S45C material will increase the weight of the motor to a certain extent and produce eddy current loss. How to further improve the inductance value through the method presented in this paper will be carried out, and at the same time, the effective way to reduce the weight and loss of the motor will be explored in the future.

**Author Contributions:** Supervision, Y.H.; Writing, review and editing, C.C. All authors have read and agreed to the published version of the manuscript.

**Funding:** This research received no external funding.

**Acknowledgments:** This work was supported in part by the Natural Science Foundation of China (51777034).

**Conflicts of Interest:** The authors declare no conflict of interest.

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
