# Peer review of "Analysis and Verification of the Method of Improving Inductance by Magnetic Endcaps in Slotless Permanent Magnet Motor"

_machines, doi:10.3390/machines10040274_

Round 1
Reviewer 1 Report
see enclosed file

Author Response
Question 1: State of the arts hold be improved and widened. To give an example: Object of research shown in Fig. 1 presents one design of a slotless machine with a very large airgap as magnetic cicuit is one side open. In many applications magnetic circiut is closed either via a back-iron, which increases inductance significantly or in a sandwich design via a second rotor with magnets. This limits airgap and leads to significantly increase values of B-field induced by permanent magnets and delivers consequently higher torque and higher efficiency. See popular motors of Faulhaber and Maxon.
Response: Under the reminding, the author realized that there were some inappropriate descriptions in the second paragraph of Introduction. This part of the content has been amended and marked.
Question 2:Replacing aluminum by steel increases weight of endcaps significantly. This counteracts the low weight advantage that slotless motors usually offer. Please give a comparison of weight.
Response: The weight of the end cap with different materials should be indicated. See section 4.1 for details.
Question 3: To understand results in Fig. 11a) it is importants to know used PWM scheme and frequency. Please discuss effect of chosen PWM scheme on PWM ripple.
Response: The motor driver adopts SPWM modulation, and the switching frequency is 10kHz. In addition, there is a measurement error in the current waveform in the original figure 11, which is now processed as shown in Figure 13.
Question 4: In line 100 Is hast o written with a subscript.
Response: The formula (3) has been Modified.

Reviewer 2 Report
Development of new types of electric machines is interesting to readers. Slotless electric machines are known to have no cogging torque and low torque ripple. However, some issues must be clarified before publishing the paper.
1) To clarify the motor design, please add some additional figures such as blow-up view of the motor.
2) Which material is used in the stator core? Is it laminated steel or soft magnetic composite?
3) There are approaches to decreasing torque ripple in conventional slot motors. The motor must be compared with a conventional PM motor of similar power and speed: their specific powers, efficiencies, etc.
4) Fig. 11b gives little information and, perhaps, can be removed. Indeed, the spectrum is cut at frequency much lower than PWM one.
5) Which are the efficiencies and the PWM losses in winding in the cases of Al and ferromagnetic end cups?
Author Response
Question 1: To clarify the motor design, please add some additional figures such as blow-up view of the motor.
Response: The blow-up view of the prototype is shown in fig. 1.
Question 2: Which material is used in the stator core? Is it laminated steel or soft magnetic composite?
Response:The stator core is made of 0.2mm WTG200 laminated, and the skeleton is processed at the outer diameter.
Question 3: There are approaches to decreasing torque ripple in conventional slot motors. The motor must be compared with a conventional PM motor of similar power and speed: their specific powers, efficiencies, etc.
Response: I'm sorry that the introduction of this paper may have caused you to misunderstand, and the introduction has been revised. The introduction is only used to explain the advantages and disadvantages of slotless motor and slotted structure motor, among which the small torque ripple is one of the advantages of slotless motor. The main content of this paper is to discuss a method to improve the inductance of slotless motor with gramme winding. Of course, the torque ripple is an important research content, but it is not within the scope of this paper.
Question 4: Fig. 11b gives little information and, perhaps, can be removed. Indeed, the spectrum is cut at frequency much lower than PWM one.
Response: Original fig.11b has been removed and the measurement error in original fig. 11a has been eliminated.
Question 5: Which are the efficiencies and the PWM losses in winding in the cases of Al and ferromagnetic end cups?
There will be obvious eddy current loss in the magnetic guide end cover, the specific content has been added in the paper. Therefore, the loss of this part should be considered in the following study.

Reviewer 3 Report
Your idea and the common thread within the publication is well recognisable.
Small improvements are needed in several places:
- What material are the skeletons made of?
- Line 108: Omega is not the electrical frequency; it's the angular frequency
- Line 109: Phase peak current: The "s" must be subscript
- Unfortunately, many line numbers are missing between lines 129 and 130. Exactly at this point there is a paragraph that is duplicated: "In AC state ...".
- Please check if the dot on the letters for current and voltage is internationally common to describe complex quantities.
- Line 134, 135: Blanks are missing before the physical unit
- The quality of the figures 8 is not very good
- Figure 10 should be placed under figure 9
In the heading, all words are written in lower case. This should be revised.
Experimental Verification: Further measurements would have been helpful. E.g. measuring the efficiency at nominal operation and at partial load.
Conclusion: The summary is too short. What conclusions do you draw from the experiment? What advantages do you expect? What further work will follow?
Overall, the language should be reviewed again.
Author Response
Question 1: What material are the skeletons made of?
Response: They are made of iron skeletons, which are integrated with the stator.
Question 2: Line 108: Omega is not the electrical frequency; it's the angular frequency.
Line 109: Phase peak current: The "s" must be subscript.
Please check if the dot on the letters for current and voltage is internationally common to describe complex quantities.
Response: Corresponding modifications have been made.
Question 3: Unfortunately, many line numbers are missing between lines 129 and 130. Exactly at this point there is a paragraph that is duplicated: "In AC state ...".
Response: The author may accidentally change the formatting of this part of the template during the writing process and may need the help of the editor to adjust it.
Question 4: Line 134, 135: Blanks are missing before the physical unit
Response: Corresponding modifications have been made.
Question 5: The quality of the figures 8 is not very good.
Response: Original Figure 8 has been redrawn, see Figure 10 for details.
Question 6: Figure 10 should be placed under figure 9.
Response: Corresponding modifications have been made.
Question 7:In the heading, all words are written in lower case. This should be revised.
Response: Corresponding modifications have been made
Question 8: Experimental Verification: Further measurements would have been helpful. E.g. measuring the efficiency at nominal operation and at partial load.
Response: In this paper, the analysis content is verified by testing the inductance parameters under different conditions and the load current obtained by sampling, which can effectively verify the theoretical analysis. Of course, the comprehensive test of the motor can show the performance of the prototype very well. Unfortunately, there is no power analyzer to test the efficiency of prototype.
Question 9: Conclusion: The summary is too short. What conclusions do you draw from the experiment? What advantages do you expect? What further work will follow?
Response: Corresponding modifications have been made according to the review comments.

Round 2
Reviewer 2 Report
It is expected that the proposed motor has lower specific characteristic compared to conventional toothed motors with the same quantity of active materials due to long nonmagnetic gap between the rotor and the stator core, including the air gap and the winding. On other hand, the obvious advantage of the motor is the absence of the cogging torque. So, the compromise leading to choosing the motor of the proposed design or the conventional one is not revealed sufficiently. Since the paper proposes a significantly new motor design, it can be recommended to the publication with some doubts.